# SIGMark: Scalable In-Generation Watermark with Blind Extraction for Video Diffusion

Xinjie Zhu*
Lenovo Research, Beijing, China
zhuxj11@lenovo.com

Zijing Zhao*
Lenovo Research, Beijing, China
zhaozj23@lenovo.com

Hui Jin
Lenovo Research, Beijing, China
jinhui8@lenovo.com

Qingxiao Guo
Lenovo Research, Beijing, China
guoqx2@lenovo.com

Yilong Ma
Lenovo Research, Beijing, China
mayl9@lenovo.com

Yunhao Wang
Lenovo Research, Beijing, China
wangyh43@lenovo.com

Xiaobing Guo
Lenovo Research, Beijing, China
guoxba@lenovo.com

Weifeng Zhang†
Lenovo Research, Beijing, China
weifengz@lenovo.com

## Abstract

Artificial Intelligence Generated Content (AIGC), particularly video generation with diffusion models, has been advanced rapidly. Invisible watermarking is a key technology for protecting AI-generated videos and tracing harmful content, and thus plays a crucial role in AI safety. Beyond post-processing watermarks which inevitably degrade video quality, recent studies have proposed distortion-free in-generation watermarking for video diffusion models. However, existing in-generation approaches are non-blind: they require maintaining all the message-key pairs and performing template-based matching during extraction, which incurs prohibitive computational costs at scale. Moreover, when applied to modern video diffusion models with causal 3D Variational Autoencoders (VAEs), their robustness against temporal disturbance becomes extremely weak. To overcome these challenges, we propose SIGMark, a Scalable In-Generation watermarking framework with blind extraction for video diffusion. To achieve blind-extraction, we propose to generate watermarked initial noise using a Global set of Framewise PseudoRandom Coding keys (GF-PRC), reducing the cost of storing large-scale information while preserving noise distribution and diversity for distortion-free watermarking. To enhance robustness, we further design a Segment Group-Ordering module (SGO) tailored to causal 3D VAEs, ensuring robust watermark inversion during extraction under temporal disturbance. Comprehensive experiments on modern diffusion models show that SIGMark achieves very high bit-accuracy during extraction under both temporal and spatial disturbances with minimal overhead, demonstrating its scalability and robustness. Our code is available at https://github.com/JeremyZhao1998/SIGMark-release.

## 1 Introduction

In the field of Artificial Intelligence Generated Content (AIGC), diffusion models have rapidly advanced image and video generation (Croitoru et al., 2023; Cao et al., 2024). Latent diffusion models proposed by Rombach et al. (2022) generates images by denoising sampled noise in latent space. Extending from this, video diffusion models generate temporally coherent frame sequences by enforcing both spatial and temporal consistency (Ho et al., 2022). With the rapid proliferation of AI-generated videos, privacy and security concerns have become increasingly critical (Wang et al., 2024). On the one hand, as a widely used creative tool, AI-generated high-quality videos constitute valuable intellectual property (IP) and necessitate reliable copyright identification. On the

---

*Xinjie Zhu and Zijing Zhao contributed equally to this work.
†Corresponding author: Weifeng Zhang (weifengz@lenovo.com).

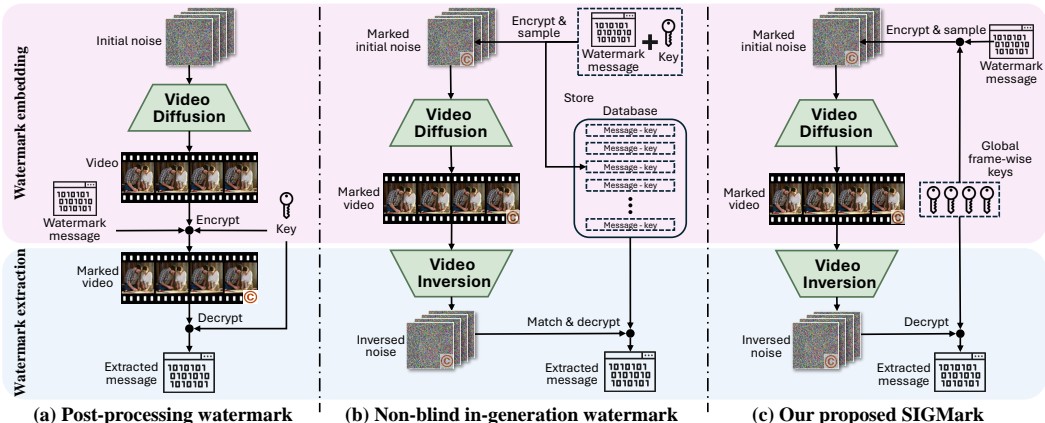

Figure 1: (a) Post-processing watermarks: embedding watermarks in pixel-space which inevitably degrades video quality. (b) Existing in-generation methods: maintaining all the message-key pairs for matching, incurring high extraction costs and poor robustness. (c) Our proposed SIGMark: a blind watermarking framework with global frame-wise PRC keys which is inherently scalable.

other hand, the ease of producing harmful or misleading content calls for strict control mechanisms, requiring effective methods to trace their source of generation.

To meet the demands of privacy and security, watermarking technology (Cox et al., 2007) has been widely applied in AIGC (Luo et al., 2025). Invisible watermarking embeds information in a way imperceptible to the human eye, thereby preserving visual quality while remaining robust to various distortions and even malicious attacks (Wang et al., 2023). For AI-generated videos, a straightforward approach is to treat them like conventional videos and apply invisible watermarking to each frame after generation (Luo et al., 2023; Zhang et al., 2024), known as post-processing watermarking (see Figure 1(a)). However, such methods inevitably introduce redundant information, thereby degrading overall video quality. Recently, in-generation watermarking has been explored for both image (Yang et al., 2024; Li et al., 2025) and video generation (Hu et al., 2025a;b). As illustrated in Figure 1(b), these methods embed watermark messages during generation process, typically by sampling a watermarked initial noise in which the message is encoded using a secret key. During extraction, the watermarked video is inverted (commonly via Denoising Diffusion Implicit Models (DDIM) inversion) back into latent noise, and then decoded with the key to recover the watermark. These approaches have been theoretically proven to be distortion-free for diffusion models.

Although in-generation video watermarking offers the advantage of being distortion-free, it still faces two critical challenges: (1) *High extraction cost at scale.* When extracting watermark messages from videos subject to temporal disturbances (e.g., frame removal during compression), current methods rely on template matching with the original latent noise. This requires maintaining all message-key pairs during watermark embedding, and computing matching functions across the entire database, as shown in Figure 1(b). Such approaches are non-blind, with the extraction cost growing linearly with the scale of users or generation requests, severely limiting scalability. (2) *Poor temporal robustness.* Modern video diffusion models (Yang et al., 2025; Kong et al., 2024) employ causal 3D Variational Autoencoders (VAE), which decode a group of adjacent frames from one temporal dimension of latent features. During extraction, video inversion requires the correct grouping of frames to reconstruct the latent feature. Temporal disturbance which disrupts the grouping will produce unrelated latent features, ultimately leading to very low extracted bit accuracy.

To address these issues and ensure usability at large-scale video generation platforms, we introduce SIGMark: scalable in-generation watermarking with blind extraction for video diffusion models.

To reduce the *high extraction cost at scale*, we propose Global Frame-wise PseudoRandom Coding (GF-PRC) scheme for watermark embedding, thereby enabling a blind watermarking framework. Specifically, a global set of frame-wise PRC keys (pseudorandom error-correction code by Christ & Gunn (2024)) is shared across all generation requests, with each key assigned to a group of temporal frames. Watermark messages are encoded into random latent noise with these keys, which preserves

diversity and generative performance. During extraction, the global frame-wise PRC directly decodes the inverted noise without matching with the original messages. Throughout this process, the system only needs to maintain the global frame-wise PRC keys, reducing extraction complexity from linear in the number of generation requests to constant, and achieving strong scalability.

To enhance *temporal robustness*, we introduce a Segment Group-Ordering (SGO) module tailored to causal 3D VAEs in modern video diffusion models. Specifically, for a video potentially affected by temporal disturbances, we first partition it into motion-consistent segments using Farnebäck optical flow; within each segment, a sliding-window grouping detector infers the original causal frame groups. This procedure recovers the correct grouping and, in turn, yields accurate inverted latents for watermark extraction under temporal disturbances.

We conduct comprehensive experiments on modern video diffusion models (HunyuanVideo by Kong et al. (2024) and Wan-2.2 by Wan et al. (2025)), covering both text-to-video (T2V) and image-to-video (I2V) pipelines. A subset of VBench-2.0(Zheng et al., 2025) under 18 evaluation dimensions is sampled to generate 400 videos for evaluation. Results show that our method achieves very high bit accuracy with high watermark capacity with minimal extraction cost. Our method maintains high accuracy under spatial and temporal disturbances, demonstrating strong robustness. Our code is available at https://github.com/JeremyZhao1998/SIGMark-release.

In conclusion, the main contributions of this paper are:

- We identify two critical issues in existing in-generation video watermarks: high extraction cost and poor temporal robustness, which hinder their scalability to large platforms.
- We propose SIGMark, a scalable in-generation watermarking framework with blind extraction for video diffusion, effectively addressing the limitations of scalability and robustness.
- We conduct extensive experiments for SIGMark on modern video diffusion models and comprehensive evaluation benchmarks, demonstrating its effectiveness and robustness.

## 2 RELATED WORKS

### 2.1 DIFFUSION MODELS

In the field of Artificial Intelligence Generated Content (AIGC), diffusion models have rapidly advanced image and video generation. Latent diffusion model (LDM)(Rombach et al., 2022) synthesizes content by denoising sampled noise in latent space and decoding it through a Variational Autoencoder (VAE). Building on LDM, works such as SDXL (Podell et al., 2024), ControlNet (Zhang et al., 2023) and DiT (Peebles & Xie, 2023) further improve image generation with stronger photorealism and higher resolution, fine-grained structural control, and greater scalability. Extending from images, video diffusion models tackle the task of generating temporally coherent frame sequences (Ho et al., 2022). Following the success of Sora (OpenAI, 2024), a wave of open-source video diffusion models including KLING (Kuaishou Technology, 2024), HunyuanVideo (Kong et al., 2024), and Wan (Wan et al., 2025) has emerged. They adopt causal 3D VAEs that compress videos along spatial and temporal dimensions to form compact latent sequences for diffusion, enabling longer, higher-quality videos with improved temporal consistency. Our research focuses on watermarking for diffusion-generated videos and evaluates on HunyuanVideo and Wan-2.2.

### 2.2 VIDEO WATERMARKING

Video invisible watermarking embeds imperceptible, durable signals in video to enable rights management and piracy deterrence (Asikuzzaman & Pickering, 2017; Aberna & Agilandeeswari, 2024). The straightforward image-based watermarking approaches operate frames individually (Hartung & Girod, 1998b; Hernandez et al., 2000), while video-based works explicitly exploit temporal information via compressed domain (Biswas et al., 2005; Noorkami & Mersereau, 2007) and motion vectors produced during compression (Mohaghegh & Fatemi, 2008). Recently, deep learning has been applied to watermarking for images (Zhu et al., 2018) and videos (Ben Jabra & Ben Farah, 2024): Zhang et al. (2019) introduced RivaGAN, training an encoder-decoder framework for robust watermarking. Subsequent work further advances robustness and capacity through curriculum learning (Ke et al., 2022), low-order recursive Zernike-moment embedding (He et al., 2023), multiscale

distribution modeling (Luo et al., 2023), complex wavelet transforms (Yasen et al., 2025; Huang et al., 2025), adversarially optimization under frequency domain Huang et al. (2024a), and spatiotemporal attention (Li et al., 2024; Yan et al., 2025). However, embedding extra signals inevitably degrades visual quality. We instead pursue in-generation watermarking for diffusion models, which is training-free and provably distortion-free.

## 2.3 IN-GENERATION WATERMARKING FOR DIFFUSION MODELS

With the rapid progress of image and video diffusion models, watermarking for diffusion-generated content has likewise gained traction. Recent work integrates watermark embedding into the generative process to reduce performance degradation. Fernandez et al. (2023) fine-tune the LDM decoder using a pre-trained watermark extractor, enabling reliable extraction from images produced by the fine-tuned model. Yang et al. (2024) introduce Gaussian Shading, the first approach to sample watermarked initial noise for image generation, which is provably distortion-free. Subsequent studies further enhance robustness during the embedding and inversion phases (Li et al., 2025; Fang et al., 2025). In-generation watermarking has also been extended from images to videos: Liu et al. (2025) propose a two-stage implanting scheme during the diffusion process. Other works encrypt the watermark message into the initial latent noise via Gaussian sampling (Hu et al., 2025a) or dynamic tree-ring (Zeng et al., 2025), preserving video generation quality. Hu et al. (2025b) adopt PRC for watermark encryption and decryption, maintaining generative diversity across messages. Despite being distortion-free, advanced in-generation methods by Hu et al. (2025a;b) still face high extraction costs at scale and poor temporal robustness. We are the first to identify and address these issues in modern video diffusion models, achieving strong scalability and temporal robustness.

## 3 METHOD

### 3.1 PROBLEM FORMULATION

This paper focuses on in-generation watermarking for video diffusion models. We first formalize the problem. Given a diffusion model $\mathcal{M}$, our goal is to embed a watermark message $m$ into generated video frames $\text{VF}(m) = \text{Embed}(\mathcal{M}; m)$ without degrading the performance of the diffusion model. The watermarked video frames may undergo temporal disturbances (e.g., frame drops) or spatial disturbances (e.g., cropping), yet the tampered frames $\text{VF}'$ should still permit reliable watermark extraction. We adopt the blind-watermark setting: during extraction, neither the original message $m$ nor the original generated video frames VF is available, and the message is recovered solely from the tampered video frames and the model, i.e., $\hat{m} = \text{Extract}(\mathcal{M}; \text{VF}')$. We emphasize robustness: even under strong disturbances to $\text{VF}'$, the recovered message $\hat{m}$ remains close to $m$.

During watermark embedding, we follow the in-generation scheme of Yang et al. (2024); Hu et al. (2025a). The watermark message $m$ is encrypted into the initial latent noise without altering its distribution, i.e., $z_0(m) \sim \mathcal{N}(0, \mathbf{I})$. The model then denoises this noise with text prompts to generate videos, thereby preserving generative performance. However, existing methods require the original message $m$ for template matching during extraction, limiting scalability in real-world deployments. Moreover, when applied to modern diffusion models with causal 3D VAEs, they exhibit poor robustness under temporal disturbances.

### 3.2 FRAMEWORK OVERVIEW

An overview of our scalable in-generation watermarking framework, SIGMark, is shown in Figure 2. We follow the in-generation watermarking scheme which embeds the watermark message into the initial latent noise to generate distortion-free watermarked video through diffusion (left part of Figure 2), and then process the video through inversion to obtain inverted latent noise for watermark extraction (right part of Figure 2). To enable blind watermarking and reduce extraction cost at scale, we introduce a Global Frame-wise Pseudo-Random Coding (GF-PRC) scheme for message encryption and decryption. During embedding, the watermark message is encoded into the initial latent noise using a global set of frame-wise PRC keys, each key assigned to one temporal dimension of latent features. During extraction, given a possibly tampered video, a Segment Group-Ordering (SGO) module restores the correct causal frame grouping by Farnebäck optical flow segmentation

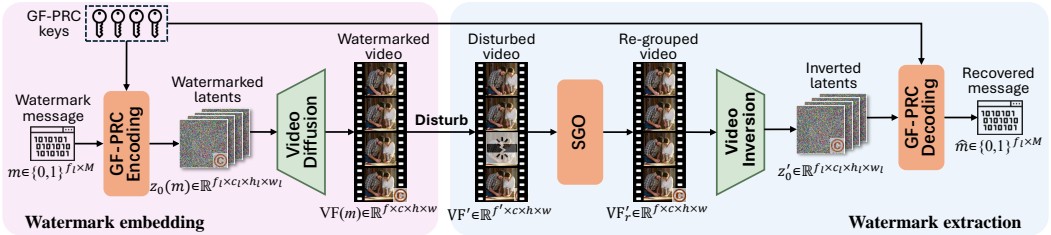

Figure 2: Overview of our proposed SIGMark. Embedding: We encode the watermark message into the initial latent noise using a Global set of Frame-wise Pseudo-Random Coding (GF-PRC) keys. The diffusion model then denoises this noise into video frames that carry the embedded messages. Extraction: A (possibly disturbed) video is first processed by our proposed Segment Group-Ordering (SGO) module to recover the correct causal frame grouping, then inverted to obtain the latent noise, from which the message is decoded using the GF-PRC keys. The system stores only the GF-PRC keys for both embedding and extraction, enabling blind watermarking.

and sliding-window grouping detector. We then perform inversion to recover the watermarked latent and decode the message with the GF-PRC keys. GF-PRC enables blind extraction where only the global keys are stored in the system, while maintaining high-accuracy message recovery under disturbances. We elaborate the details of our proposed modules in the following sections.

## 3.3 WATERMARK EMBEDDING

Following the in-generation watermarking scheme(Yang et al., 2024; Hu et al., 2025a), we map the watermark message bits into the initial latent noise without affecting the Gaussian distribution of the noise for watermark embedding. To achieve blind watermarking, we propose to utilize a Global set of Frame-wise Pseudo-Random coding (GF-PRC) scheme for watermark embedding.

### 3.3.1 VIDEO GENERATION BY MODERN DIFFUSION MODELS

For a modern video diffusion model(Kong et al., 2024; Wan et al., 2025) $\mathcal{M}$, which consists of a text prompt encoder $E_{\text{text}}$, a denoising diffusion transformer $T$, and the encoder $E_{\text{3D}}$ and decoder $D_{\text{3D}}$ of the causal 3D Variational Autoencoders (VAE), the denoising steps happen on latent-space features $z \in \mathbb{R}^{f_l \times c_l \times h_l \times w_l}$, where $f_l, c_l, h_l, w_l$ denote the frame, channel, height, and width dimensions in latent space. The diffusion transformer $T$ denoises a randomly initialized latent noise $z_0 \sim \mathcal{N}(o, \mathbf{I})$ guided by the text prompt: $z_\tau = \text{Denoise}\big(T; z_o; E_{\text{text}}(\text{prompt})\big)$, where $\tau$ denotes the number of denoising steps. The denoised latent feature $z_\tau$ is then processed by the causal 3D VAE decoder to generate video frames $\text{VF} \in \mathbb{R}^{f \times c \times h \times w}$, where $f, c, h, w$ denote the frame, channel, height, and width of the generated video. The whole process can be formulated as:

$$\text{VF} = \text{Diffusion}(\mathcal{M}; z_o; \text{prompt}) \tag{1}$$

$$= D_{\text{3D}}\Big(\text{Denoise}\big(T; z_o; E_{\text{text}}(\text{prompt})\big)\Big). \tag{2}$$

The causal 3D VAE introduces information compression along both spatial and temporal dimensions, where $f = f_l \times d_t$, $h = h_l \times d_s$, and $w = w_l \times d_s$, with $d_t, d_s$ denoting the temporal and spatial compression ratios, respectively. As a result, a group of $d_t$ frames is decoded from one temporal dimension of the latent features. Our proposed SIGMark embeds the watermark message into the latent noise via the GF-PRC scheme, yielding a sequence of watermark bits carried by a causal group of video frames, as detailed in the next paragraph.

### 3.3.2 GLOBAL FRAME-WISE PSEUDORANDOM CODING SCHEME

We propose embedding all watermark messages using a global set of frame-wise pseudorandom coding keys, as shown in Figure 3. Specifically, for a watermark message $m \in \{0, 1\}^{f_l \times M}$ where $f_l$ is the number of latent frames and $M$ is the bit length carried by each causal frame group, we encode $m$ into a random template bit sequence $\text{TP} \in \{0, 1\}^{f_l \times (c_l \times h_l \times w_l)}$ using the pseudorandom

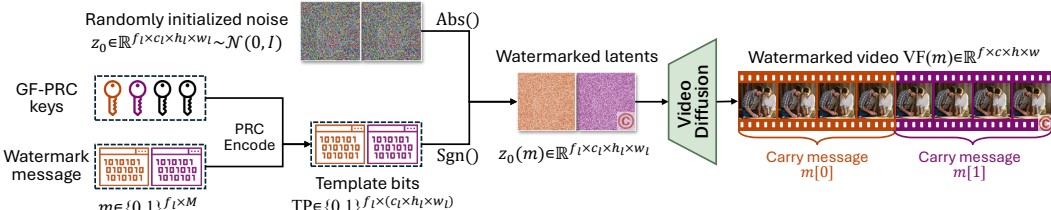

Figure 3: Watermark embedding with GF-PRC scheme. We set $f_l = 2, f = 8$ with compression ratio $d_t = f/f_l = 4$ as an example. Orange and purple denote message $m[0]$ and $m[1]$ respectively.

error-correction code (simplified as pseudorandom code, PRC) proposed by Christ & Gunn (2024):

$$\text{TP}[i] = \text{PRC.Encode}\big(m[i]; K[i]\big), \quad i \in \{0, 1, 2, ..., f_l - 1\} \tag{3}$$

Here, $K$ denotes the pseudorandom error-correction coding keys; $K[i]$ is the key for frame dimension $i$ in latent space, with $m[i] \in \{0,1\}^M$ and $\text{TP}[i] \in \{0,1\}^{(c_l \times h_l \times w_l)}$. We allocate one PRC key per causal frame group in latent space, enhancing robustness and enabling causal frame grouping and ordering information recovery (detailed in the next section). Given the randomized template TP, we map the watermark message into the initial latent noise by element-wise modulation:

$$z_0(m) = (\text{TP} * 2 - 1) * |z_0| \tag{4}$$

With the random absolute value from Gaussian sampling and the randomized template bits as the modulation signal, the embedded initial noise remains Gaussian, $z_0(m) \sim \mathcal{N}(0, 1)$, and thus does not degrade the diffusion model's generative performance. Consequently, the generated video frames are watermarked with $m$ via:

$$\text{VF}(m) = \text{Diffusion}\big(\mathcal{M}; z_0(m); \text{prompt}\big) \tag{5}$$

Note that in our GF-PRC scheme, the frame-wise PRC keys are global: every generation request shares the same key set, and each latent frame dimension carries one watermark sequence encoded by its corresponding PRC key. The total number of GF-PRC keys can be set to the maximum frame capacity of the video generation system, enabling watermarking for videos of arbitrary length. PRC by Christ & Gunn (2024) introduces a pseudo-random mapping that can encode even the same message into different random template bits, thereby preserving randomness in the initial latent noise under global keys, which traditional stream ciphers (e.g., ChaCha20(Bernstein et al., 2008)) used in prior in-generation non-blind methods(Yang et al., 2024; Hu et al., 2025a) cannot provide with the fixed keying material. The detailed explanation can be seen in Appendix A.

## 3.4 WATERMARK EXTRACTION

As shown in Figure 2, given a test video, we first apply the Segment Group-Ordering module to recover the grouping and ordering information of video frames, enhancing robustness against temporal disturbances, then perform diffusion-based inversion to obtain the inverted latent noise, and finally apply PRC decoding to recover the message.

### 3.4.1 SEGMENT GROUP-ORDERING MODULE

A watermarked video may encounter various disturbances, among which temporal disturbances are particularly critical for modern diffusion models with causal 3D VAEs. As shown on the left of Figure 4, the original frames $\text{VF}(m)$ are generated in causal groups of 4 (frames sharing the same color). After video clipping or frame drops, the disturbed video $\text{VF}'$ loses both grouping and ordering information. If such frames are fed directly into the causal 3D VAE encoder, the mis-grouping (e.g., differently colored frames within a group) produces latent features that are inconsistent with those of $\text{VF}(m)$, thereby preventing reliable recovery of the message bits via inversion. To address the issue of weak temporal robustness, we propose a Segment Group-Ordering (SGO) module as shown in the right part of Figure 4 to recover the grouping and ordering information of the disturbed video for correct encoding.

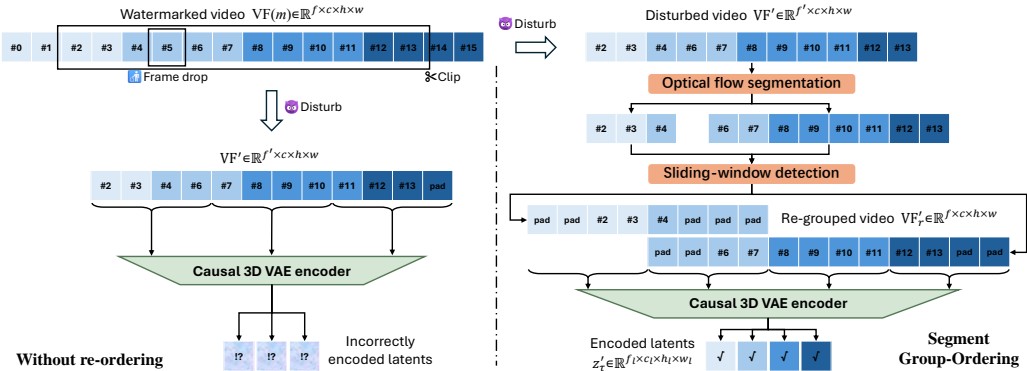

Figure 4: Segment Group-Ordering (SGO) module. We set compression ratio $d_t = f/f_l = 4$ as an example. When temporal disturbances (e.g., clipping or frame drops) occur, the causal grouping is disrupted; without re-ordering, this leads to incorrectly encoded latent features. Our SGO module restores the correct grouping and ordering, yielding robust latent features for video inversion.

**Optical flow segmentation:** We propose a optical-flow method to partition a video into maximally contiguous subsequences with consistent temporal dynamics. For each adjacent pair $(\text{VF}[t], \text{VF}[t+1])$, we compute bidirectional Farnebäck flow and derive three indicators of temporal consistency: (i) median flow magnitude, (ii) forward-backward consistency, and (iii) motion-compensated residual. These are normalized via median absolute deviation and combined with a weighted sum to form a discontinuity score. The score is smoothed with a Gaussian filter, and temporal cut points are detected using hysteresis thresholding. The procedure runs near real time and outputs contiguous frame segments. Within each segment, we then perform sliding-window grouping detection. Further details of the optical-flow segmentation algorithm are provided in Appendix C.

**Sliding-window detection:** Given a contiguous frame segment, we only need to identify the first frame of a causal group, and the subsequent frames can then be grouped correctly. To this end, we leverage the global frame-wise PRC keys. Specifically, we pad $(d_t - 1)$ frames at the beginning of the segment and apply a sliding window over the padded sequence. At sliding index $j$, we invert frames $[j : j + 2 * d_t]$ to obtain two latent frame dimensions $z_0'[j]$ and $z_0'[j+1]$. Using the global PRC keys, the frame index of each latent can be determined by:

$$\hat{\text{Idx}}[j] = \text{argmax}\big(\text{PRC.Detect}(z_0'[j]; K[0, 1, ..., f_l])\big), \hat{\text{Idx}}[j] \in \{0, 1, ..., f_l\} \tag{6}$$

We stop sliding when the detections are consecutive, i.e., $\hat{\text{Idx}}[j] + 1 = \hat{\text{Idx}}[\hat{j} + 1]$, indicating that the current grouping yields the correct segment start for inversion. The results from all segments are then merged to produce the re-grouped frames $\text{VF}_r'$, as shown in Figure 4. Note that empty slots within each group are filled preferentially with available frames; any remaining slots are then padded. If an entire group contains no frames (caused by frame drop), we assign padding based on the nearest available frame. This procedure recovers correct ordering and grouping information and is robust to frame insertion, swapping, dropping, and clipping. Further details of the sliding-window detection algorithm are provided in Appendix C.

### 3.4.2 MESSAGE BITS EXTRACTION

Following the in-generation watermarking paradigm(Yang et al., 2024; Hu et al., 2025a), we perform inversion on the re-grouped frames to extract the watermark bits. The latent features $z_\tau' \in \mathbb{R}^{f_l \times c_l \times h_l \times w_l}$ are first obtained via the causal 3D VAE encoder, and then inverted through the diffusion transformer (the flow-matching Euler discrete inversion for HunyuanVideo(Kong et al., 2024) and Wan(Wan et al., 2025)):

$$z_\tau' = E_{3D}(\text{VF}_r') \tag{7}$$

$$z_0' = \text{Inversion}(\mathcal{M}; z_\tau'; \text{prompt}_\emptyset) \tag{8}$$

where $\text{prompt}_\emptyset$ denotes an empty prompt, since no original generation information is available. Given the inverted latent noise $z_0' \in \mathbb{R}^{f_l \times c_l \times h_l \times w_l}$, we decode the GF-PRC keys $K$ by applying

Table 1: Video watermarking results: message recovery bit accuracy and video quality score.[1]

| Diffusion model | | HunyuanVideo T2V | | | | HunyuanVideo I2V | | | |
|---|---|---|---|---|---|---|---|---|---|
| Watermarking | | 512 bits | | 512×16 bits | | 512 bits | | 512×16 bits | |
| Method | Category | Bit acc↑ | V-score↑ | Bit acc↑ | V-score↑ | Bit acc↑ | V-score↑ | Bit acc↑ | V-score↑ |
| No-mark | – | – | 0.490 | – | 0.490 | – | 0.463 | – | 0.463 |
| DCT | Post | 0.889 | 0.424 | 0.862 | 0.423 | 0.890 | 0.452 | 0.858 | 0.456 |
| DT-CWT | Post | 0.619 | 0.416 | 0.650 | 0.436 | 0.627 | 0.458 | 0.611 | 0.463 |
| VideoMark | None-blind | 0.873 | 0.507 | 0.758 | 0.502 | 0.846 | 0.483 | 0.707 | 0.482 |
| VideoShield | None-blind | 1.000 | 0.497 | 0.991 | 0.506 | 1.000 | 0.482 | 0.999 | 0.482 |
| SIGMark(Ours) | Blind | 0.958 | 0.506 | 0.885 | 0.499 | 0.981 | 0.472 | 0.905 | 0.488 |

PRC to the signs of $z_0'$:

$$\hat{m}[i] = \text{PRC.Decode}\left(\frac{\text{Sgn}\left(z_0'[i]\right) + 1}{2}; K[i]\right), \quad i \in \{0, 1, 2, ..., f_l - 1\} \tag{9}$$

where $\text{Sgn}(\cdot)$ is the sign function. With the global PRC key set, no additional information needs to be stored during generation, and no template matching (as in Hu et al. (2025a;b)) is required during extraction. Therefore, our proposed SIGMark enables blind watermarking with minor computation cost and strong scalability. Details of message extraction cost are provided in Appendix B.

## 4 EXPERIMENTS

In this section, we conduct experiments on modern video diffusion models. We detail the experimental setup in Section 4.1, report bit accuracy comparisons with existing methods in Section 4.2, assess temporal and spatial robustness in Section 4.3, conduct ablation studies in Section 4.4 and present evidence of scalability in Section 4.5.

### 4.1 EXPERIMENTAL SETTINGS

#### 4.1.1 DATASETS AND METRIC

To comprehensively assess modern video diffusion models, we conduct experiments on a subset of prompts from VBench-2.0(Zheng et al., 2025), which offers more evaluation dimensions and more complex prompts than VBench-1.0(Huang et al., 2024b) used by existing researches on earlier diffusion models(Hu et al., 2025a;b). Among the 18 dimensions in VBench-2.0, we select 3 prompts for the "diversity" dimension and generate 20 videos per prompt; for the remaining 17 dimensions, we select 5 prompts each and generate 4 videos per prompt, yielding a total of 400 videos. We report bit accuracy between the embedded message $m$ and the recovered message $\hat{m}$. Video quality is evaluated using the VBench-2.0 protocols to obtain an overall quality score. Detailed examples of the constructed prompt set are provided in Appendix D.

#### 4.1.2 IMPLEMENTATION DETAILS

We evaluate on two open-source video diffusion models: HunyuanVideo(Kong et al., 2024) and Wan-2.2(Wan et al., 2025), both employing causal 3D VAEs with temporal and spatial compression ratios $d_t = 4$ and $d_s = 8$. For each model, we consider both text-to-video (T2V) and image-to-video (I2V) tasks. For T2V, we generate videos at resolution $h = 512$, $w = 512$, with 65 frames per video. The first frame is processed independently by the diffusion model, so no watermark is embedded in the first frame. The remaining $f = 64$ frames form $f_l = f/d_t = 16$ causal groups, for each group we maintain one frame-wise PRC key globally. For I2V, we use the same text prompt set to generate

---

[1]We implement DCT and DT-CWT through the open-source code of video-invisible-watermark and blind-video-watermark respectively, and we implement VideoMark and VideoShield by adapting their officially released code and hyper-parameter to new diffusion models and prompts.

Table 2: Watermark extraction bit accuracy under disturbances on HunyuanVideo I2V.

| Method | Spatial disturbance | | | | Temporal disturbance | | | |
|---|---|---|---|---|---|---|---|---|
| | w/o | G.noise | cmprs | blur | w/o | drop | insert | clip |
| VideoMark | 0.85 | $0.64_{\downarrow 0.21}$ | $0.63_{\downarrow 0.22}$ | $0.64_{\downarrow 0.21}$ | 0.71 | $0.52_{\downarrow 0.19}$ | $0.51_{\downarrow 0.20}$ | $0.51_{\downarrow 0.20}$ |
| VideoShield | 1.00 | $1.00_{\downarrow 0.00}$ | $0.99_{\downarrow 0.01}$ | $1.00_{\downarrow 0.00}$ | 0.99 | $0.89_{\downarrow 0.10}$ | $0.84_{\downarrow 0.15}$ | $0.83_{\downarrow 0.16}$ |
| SIGMark(Ours) | 0.98 | $0.89_{\downarrow 0.09}$ | $0.84_{\downarrow 0.14}$ | $0.95_{\downarrow 0.03}$ | 0.91 | $0.81_{\downarrow 0.10}$ | $0.87_{\downarrow 0.04}$ | $0.85_{\downarrow 0.06}$ |

Table 3: Ablation study on our proposed modules on HunyuanVideo I2V

| Method | Watermark embedding | | Watermark extraction | | | |
|---|---|---|---|---|---|---|
| | Single PRC | GF-PRC(Ours) | w/o SGO | w/o OF-seg | w/o SW-det | SGO(Ours) |
| Bit acc | 0.707 | 0.905 | 0.534 | 0.762 | 0.823 | 0.869 |

the initial image prompt by FLUX(Black Forest Labs et al., 2025) for evaluation. The generated image prompt examples are shown in Appendix D.

## 4.2 WATERMARK EXTRACTION RESULTS

In this section, we compare our watermarking performance with existing approaches. As presented in Table 1, we report message bit accuracy ("Bit Acc") and the VBench-2.0 score ("V-score") for HunyuanVideo. Additional results for Wan-2.2 are provided in Appendix D. We consider two configurations: (1) embedding an identical 512-bit watermark message in every causal frame group of a video, and (2) embedding a distinct 512-bit watermark message per causal frame group, corresponding to total capacities of 512 bits and 512×16=8192 bits per video, respectively.

We compare our method against four baselines: DCT(Hartung & Girod, 1998a), DT-CWT(Coria et al., 2008), VideoShield(Hu et al., 2025a), and VideoMark(Hu et al., 2025b). DCT and DT-CWT are widely used post-processing watermarking methods for general videos, whereas VideoShield and VideoMark are recent in-generation methods for video diffusion models but are non-blind. For post-processing methods, we first generate a set of standard videos with the diffusion model and then apply watermarking of different settings. As shown in Table 1, post-processing watermarking causes notable quality degradation relative to no-watermark videos, while in-generation methods leave visual quality essentially unaffected. Our proposed SIGMark can be proved to be performance-lossless. Detailed proof can be found in Appendix A. Our proposed SIGMark with blind extraction attains very high bit accuracy under both lower and higher capacity settings, surpassing VideoMark by large margins and remaining competitive with VideoShield which requires access to the original watermark information, thereby demonstrating the effectiveness of our approach.

## 4.3 ROBUSTNESS

We assess the robustness of watermarking methods under both spatial and temporal perturbations in Table 2. We apply spatial disturbance under 512 bits watermark and temporal disturbances under 512x16 bits. "w/o" denotes without disturbance. "G.noise", "cmprs" and "blur" denotes Gaussian noise, mpeg compression and blurring, respectively. In temporal disturbances, we randomly drop, insert or clip out 30 frames. The performance degradation compared with no disturbances is marked as subscript. For spatial disturbances, our approach incurs only marginal performance degradation, particularly when compared to VideoMark. Under the challenging temporal disturbances, VideoMark and VideoShield suffer substantial degradation on diffusion models with causal 3D VAEs, primarily due to incorrect grouping of causal frame groups. Our method mitigates this issue, achieving negligible performance loss and thereby improving temporal robustness. It is worth noting that our method does not attain 100% bit accuracy. We attribute this to the relationship of error-tolerance characteristics of PRC coding and the accuracy of diffusion inversion for different models. A detailed analysis of PRC coding robustness is provided in Appendix E.

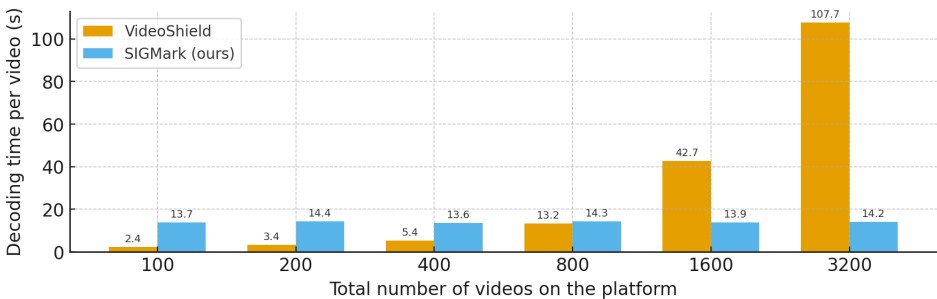

Figure 5: The decoding time cost during watermark extraction.

## 4.4 ABLATION STUDIES

We conduct ablation studies to evaluate the contribution of each proposed component in Table 3. "OF-seg" and "SW-det" denotes optical flow segmentation and sliding window detection, respectively. For the ablation study of watermark embedding modules, we evaluate a 512×16-bit watermark on generated videos without disturbances. For the ablation study of watermark extraction modules, we evaluate a 512×16-bit watermark under temporal disturbance by inserting 30 random frames. When SGO and OF-seg are removed, we simply truncate the video to the target length. When SW-det is removed, we perform PRC detection by assuming that each segment starts at the beginning of a frame group. For the GF-PRC scheme, removing it reduces our method to the same coding strategy as VideoMark, which exhibits limited bit accuracy under the blind setting without template matching. In contrast, GF-PRC not only enables blind extraction but also improves bit accuracy by introducing inter-frame redundant error tolerance. For the SGO module, both optical-flow-based segmentation and sliding-window detection are critical for recovering frame grouping and ordering; omitting either component leads to a noticeable accuracy drop.

## 4.5 EVIDENCE OF SCALABILITY

Baseline methods such as VideoShield and VideoMark are non-blind watermarking schemes: they require storing all watermark-related information (messages, encoding keys, etc.) during generation and matching against all stored information during extraction. Therefore, extraction cost grows with the total number of generated videos. Our method, SIGMark, is blind: it maintains only a global set of frame-wise PRC keys and does not require any sample-specific metadata during extraction. As a result, it supports large-scale video generation platforms with constant extraction cost. We analyze extraction time cost under scenarios where the total number of videos generated by the platform varies. Experiments are conducted under HunyuanVideo I2V with 512x16-bit watermark without disturbances. For fairness, we run inversion on GPU, and all remaining extraction steps including decryption and message matching on CPU. As shown in Figure 5, the results demonstrate that the time cost of VideoShield scales linearly with the number of generated videos, which becomes impractical as the platform scales to millions of videos. In contrast, SIGMark remains constant, demonstrating strong scalability.

## 5 CONCLUSION

Watermarking for video diffusion models is critical for ensuring safety and privacy control in AIGC. In this work, we introduce SIGMark, the first blind in-generation video watermarking method for modern diffusion models, offering strong scalability and practical applicability. To enable blind extraction without storing large-scale watermarking references, we propose a Global Frame-wise Pseudo-Random Code (GF-PRC) scheme, which encodes watermark messages into the initial latent noise without compromising video quality or diversity. To further enhance temporal robustness, we design a Segment Group-Ordering (SGO) module tailored for causal 3D VAEs, ensuring correct watermark inversion. Extensive experiments demonstrate that our approach achieves high bit accuracy with minimal overhead, validating its scalability and robustness.

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
