# Supplementary material of SIGMark: Scalable In-Generation Watermark with Blind Extraction for Video Diffusion

Xinjie Zhu[*]
Lenovo Research, Beijing, China
zhuxj11@lenovo.com

Zijing Zhao[*]
Lenovo Research, Beijing, China
zhaozj23@lenovo.com

Hui Jin
Lenovo Research, Beijing, China
jinhui8@lenovo.com

Qingxiao Guo
Lenovo Research, Beijing, China
guoqx2@lenovo.com

Yilong Ma
Lenovo Research, Beijing, China
mayl9@lenovo.com

Yunhao Wang
Lenovo Research, Beijing, China
wangyh43@lenovo.com

Xiaobing Guo
Lenovo Research, Beijing, China
guoxba@lenovo.com

Weifeng Zhang[†]
Lenovo Research, Beijing, China
weifengz@lenovo.com

In this material, we include the following contents as supplements of our paper: (1) an explanation of our performance-lossless watermarking in Appendix A; (2) an analysis of the computational overhead of our method in Appendix B, demonstrating the scalability of SIGMark; (3) detailed implementation of the proposed Segment Group-Ordering (SGO) module in Appendix C; (4) additional experimental details and results on video watermarking in Appendix D; (5) an analysis of the performance bottlenecks of our method, outlining future research directions in Appendix E; (6) a statement on the usage of Large Language Models (LLMs) in this paper in Appendix F.

## A  Impact of watermarking on video diffusion systems

### A.1  Stream cipher unavailable for blind watermarking

To embed a watermark message into the video diffusion model's initial latent noise while remaining performance-lossless, the message must be encoded via a cryptographic (pseudo) randomized encoder so that the induced perturbations are statistically indistinguishable from natural noise. Stream ciphers such as ChaCha20(Bernstein et al., 2008) can produce pseudorandom code streams and offer good noise resilience. However, their keystreams are deterministic given a fixed key (and nonce). Consequently, if a global key is reused and the message is fixed, the resulting codeword is also fixed, which harms generative diversity by mapping identical messages to identical latent sign patterns across runs. Preserving diversity would require a fresh per-generation secret (e.g., a new key/nonce) and thus auxiliary side information for extraction, rendering such schemes inherently none-blind and imposing substantial storage and coordination overhead at scale (tracking a large number of key-message pairs). Similar conclusion has been stated by Thietke et al. (2025). In contrast, our approach targets blind watermarking with a single or a set of global keys by adopting a PRC (Pseudorandom error-correction Code proposed by Christ & Gunn (2024)) that is explicitly randomized at encoding time: the same message admits many codewords drawn from a pseudorandom ensemble. This probabilistic encoding preserves sample diversity across generations without storing per-instance secrets, while still enabling blind extraction under global keys.

### A.2  Proof of our proposed method being performance-lossless

Prior works often observe that inserting watermark embedding modules degrades model performance when evaluated by metrics such as Peak Signal-to-Noise Ratio (PSNR) and Fréchet Inception Distance (FID), which are more suitable for post-processing methods. To assess methods that inte-

---

[*]Xinjie Zhu and Zijing Zhao contributed equally to this work.
[†]Corresponding author: Weifeng Zhang (weifengz@lenovo.com).

grate watermark embedding within the video generation process, we adopt a complexity-theoretic indistinguishability notion (inspired by cryptographic security definitions) to characterize the impact of watermarking on model performance. Concretely, consider a probabilistic game between a watermarked video $\mathrm{VF}(m)$ and a normally generated video $\mathrm{VF}$.

The watermarking method is *performance-lossless* if, for any probabilistic polynomial-time (PPT) tester (distinguisher) A, it holds that

$$|\Pr[\mathsf{A}(\mathrm{VF}(m)) = 1] - \Pr[\mathsf{A}(\mathrm{VF}) = 1]| < \mathrm{negl}(\rho)\,, \tag{10}$$

where $\rho$ is the security parameter (e.g., the length of the PRC keys $K$), and $\mathrm{negl}(\rho)$ denotes a function negligible in $\rho$ (i.e., smaller than the inverse of any polynomial in $\rho$).

We prove the claim by contradiction. Assume that $\mathrm{VF}(m)$ and $\mathrm{VF}$ are distinguishable by some PPT tester A with non-negligible advantage $\delta > 0$:

$$|\Pr[\mathsf{A}(\mathrm{VF}(m)) = 1] - \Pr[\mathsf{A}(\mathrm{VF}) = 1]| = \delta\,. \tag{11}$$

Let the entire video generation pipeline, comprising the diffusion transformer $\mathrm{T}_{\mathrm{diff}}$ and the causal 3D VAE decoder $\mathrm{D}_{\mathrm{3D}}$, start from an initial latent noise $z_0^{(\cdot)}$. Substituting the generative process into (11) yields

$$\left|\Pr\big[\mathsf{A}(\mathrm{D}_{\mathrm{3D}}(\mathrm{T}_{\mathrm{diff}}(z_0(m)))) = 1\big] - \Pr\big[\mathsf{A}(\mathrm{D}_{\mathrm{3D}}(\mathrm{T}_{\mathrm{diff}}(z_0))) = 1\big]\right| = \delta\,, \tag{12}$$

where $z_0(m)$ denotes the initial latent noise whose sign pattern is determined by the message $m$ via our PRC encoding with keys $K$ (hence forming a pseudorandom sign sequence), and $z_0$ denotes the initial noise used for standard generation, drawn from a truly random distribution (e.g., $z_0 \sim \mathcal{N}(0, \mathbf{I})$).

Since $\mathrm{D}_{\mathrm{3D}}$ and $\mathrm{T}_{\mathrm{diff}}$ are deterministic PPT algorithms, they can be treated as subroutines available to the tester. Define a new tester $\mathsf{A}_{\mathrm{gen}} = \mathsf{A} \circ \mathrm{D}_{\mathrm{3D}} \circ \mathrm{T}_{\mathrm{diff}}$. Then (12) reduces to distinguishing the initial latents:

$$|\Pr[\mathsf{A}_{\mathrm{gen}}(z_0(m)) = 1] - \Pr[\mathsf{A}_{\mathrm{gen}}(z_0) = 1]| = \delta\,. \tag{13}$$

Equation (13) asserts that a PPT algorithm can distinguish a latent tensor with a pseudorandom sign pattern ($z_0(m)$) from one with truly random signs ($z_0$) with non-negligible advantage.

However, our PRC mechanism is built upon cryptographic principles: the global keys $K$ are generated via a secure pseudorandom function (PRF). A fundamental property of a secure PRF is that its output is computationally indistinguishable from uniform randomness; consequently, the sign patterns produced by $\mathrm{PRC.Encode}(K, m)$ are computationally indistinguishable from truly random signs. This contradicts (13), which would enable distinguishing PRC-generated signs from random in PPT, violating the PRF security assumption.

Therefore, the assumption in (11) is false. It follows that no PPT tester can distinguish watermarked videos from normally generated videos with non-negligible advantage, and thus our video watermarking method is performance-lossless under the above complexity-theoretic definition.

## B  COMPUTATION OVERHEAD

### B.1  COMPUTATION COMPLEXITY ANALYSIS

Consider a large-scale video generation platform based on video diffusion models. We analyze the computational complexity of watermark embedding and extraction for our method and existing in-generation watermarking approaches (Hu et al., 2025a;b). Let $f$ denote the maximum number of frames per video, $h \times w$ the spatial resolution, $t_{\mathrm{diff}}$ the time for denoising or inversion (correlated with $f \times h \times w$), $M$ the watermark message length, $K$ the key length, and $t_{\mathrm{encrypt}}$ the encryption/decryption time (correlated with $M$ and $f \times h \times w$). Let $N$ denote the number of videos or generation requests in the system.

For in-generation watermarking, the embedding process incurs no additional spatial cost beyond video generation, and the temporal cost is a single encryption step $t_{\mathrm{encrypt}}$, giving a total cost of: $t_{\mathrm{encrypt}} + t_{\mathrm{diff}}$. However, non-blind watermarking requires storing key–message or key–template pairs for extraction, resulting in an additional spatial cost of: $O(N \times (M + K))$. For our proposed

blind watermarking, only global frame-wise keys are maintained, with spatial cost: $O(f \times K)$. For extraction, non-blind watermarking requires matching against all stored watermark information. Under temporal disturbances, matching must be performed between watermark template bits for every frame, leading to a temporal complexity of: $O(N \times f \times f \times M)$, and a total complexity of: $O(N \times f^2 \times M) + t_{\text{diff}} + t_{\text{decrypt}}$. In contrast, our proposed SIGMark compares inverted latents with all frame-wise keys, yielding a total cost of: $t_{\text{diff}} + f \times t_{\text{decrypt}}$.

In typical experiments where $N$, $M$, and $f$ are of the same order, the dominant cost is $t_{\text{diff}}$. However, in large-scale systems where $N \gg f$, the parameters $f, h, w, M, K$ can be treated as constants (e.g., $f = 64$, $h = w = 512$, $M = 512$, while $N \sim 10^8$). In this setting, the $O(N)$ spatial and temporal cost of non-blind watermarking becomes prohibitive, whereas our proposed SIGMark maintains constant complexity, demonstrating strong scalability.

## B.2 Experimented computation time and memory usage on hardware

All experiments are conducted on NVIDIA A800 GPUs with `bfloat16` precision. On Hunyuan-Video, generating a single video with $f = 65$, $h = 512$, and $w = 512$ takes approximately 4 minutes, with watermark embedding time being negligible. Processing the dataset of 400 videos requires about 1600 A800 GPU hours. For watermark extraction, PRC detection and decoding take roughly 30 seconds per sample on CPU, which can potentially be optimized through parallelization. Notably, this computation time remains constant regardless of the scale of video generation.

## C Detailed implementation of SGO

### C.1 Detailed implementation of Optical Flow segmentation

We aim to partition a sequence of frames $\{I_t\}_{t=0}^{n-1}$ into contiguous segments that are likely temporally consistent, even under frame deletions, insertions, or swaps. For pre-processing, each frame is converted to grayscale and downscaled so that its short side does not exceed a fixed size (default 288), reducing computational cost without introducing upsampling artifacts. For Optical Flow Estimation, for each adjacent pair $(I_t, I_{t+1})$, we compute forward and backward Farnebäck optical flow fields, denoted as $\mathbf{F}^{t \rightarrow t+1}$ and $\mathbf{F}^{t+1 \rightarrow t}$. We then calculate boundary features. For each boundary $t | t + 1$, we compute four robust signals:

$$M_t = \text{median}_{\mathbf{x}} \left\| \mathbf{F}^{t \rightarrow t+1}(\mathbf{x}) \right\|_2, \tag{14}$$

$$C_t = \text{median}_{\mathbf{x}} \left\| \mathbf{F}^{t \rightarrow t+1}(\mathbf{x}) + \mathbf{F}^{t+1 \rightarrow t}(\mathbf{x} + \mathbf{F}^{t \rightarrow t+1}(\mathbf{x})) \right\|_2, \tag{15}$$

$$R_t = \frac{1}{255|\Omega|} \sum_{\mathbf{x} \in \Omega} \left| \hat{I}_{t \rightarrow t+1}(\mathbf{x}) - I_{t+1}(\mathbf{x}) \right|, \tag{16}$$

$$\Delta M_t = \begin{cases} 0, & t = 0, \\ |M_t - M_{t-1}|, & t \geq 1. \end{cases} \tag{17}$$

Here, $M_t$ is the median flow magnitude, $C_t$ measures forward–backward consistency, $R_t$ is the motion-compensated residual after warping $I_t$ to $I_{t+1}$ using backward flow, and $\Delta M_t$ captures speed changes. Each signal is standardized using a robust Z-score based on the median absolute deviation (MAD):

$$z(x) = 0.6745 \cdot \frac{x - \text{median}(x)}{\text{MAD}(x) + 10^{-9}}. \tag{18}$$

The final discontinuity score for boundary $t$ is a weighted combination:

$$\text{score}_t = 0.35|z_M| + 0.30 \max(z_C, 0) + 0.25 \max(z_R, 0) + 0.10 \max(z_{\Delta M}, 0). \tag{19}$$

A light Gaussian smoothing is applied to reduce jitter. We finally apply hysteresis thresholding with high and low thresholds (`score_hi, score_lo`) to detect cut points, reducing false positives from noise. Segments are then assembled as:

$$[0, i_1], [i_1 + 1, i_2], \ldots, [i_k + 1, n - 1].$$

**Complexity.** The dominant cost is optical flow estimation for $n - 1$ frame pairs, while memory overhead is minimal. The method is CPU-friendly and robust to noise, making it suitable for large-scale video analysis.

### C.2 Details of sliding-window detection

The sliding-window detection module aims to determine both the grouping and the relative position of a given video segment within the entire generated video. Leveraging the detection capability of the GF-PRC scheme, we achieve this goal efficiently. Given a video segment $\mathrm{VF}[t : t + l]$, we first pad the segment with $(c_t - 1)$ frames at the beginning, where $c_t$ denotes the temporal compression ratio of the causal 3D VAE (e.g., $c_t = 4$). We then apply a sliding window of size $2 \times c_t$ (e.g., $8$) over the padded sequence. For each window index $j$, we perform inversion to obtain the local inverted initial noise:

$$z'_\tau[j : j + 1] = E_{3\mathrm{D}}\big(\mathrm{VF}'_r[j : j + 2 \times c_t]\big), \tag{20}$$

$$z'_0[j : j + 1] = \mathrm{Inversion}\big(\mathcal{M}; z'_\tau[j : j + 1]; \mathrm{prompt}_\emptyset\big). \tag{21}$$

Next, we use the GF-PRC keys to detect whether $z'_0[j : j + 1]$ matches any global frame index using Eq. (6) from the main paper. This approach is effective because PRC detection accuracy is significantly higher than PRC decoding, as discussed in Appendix E. We also observe that repeating the same frame $c_t$ times yields high inversion and detection accuracy, enabling the padding strategy to function correctly within the sliding window. The sliding process terminates once the detected indices form a continuous sequence, as described in the main paper. The computational complexity of this procedure is at most $O(f \times c_t)$ and at least $O(f)$, which remains constant with respect to the overall generation scale.

## D    Additional experimental results

### D.1    Our constructed prompt set

To comprehensively assess both video quality and the efficiency of the watermarking framework, we adopt VBench-2.0 (Zheng et al., 2025), a benchmark specifically designed for modern video diffusion models with stronger ability. VBench-2.0 evaluates 18 dimensions, including: `Camera Motion`, `Complex Landscape`, `Complex Plot`, `Composition`, `Diversity`, `Dynamic Attribute`, `Dynamic Spatial Relationship`, `Human Anatomy`, `Human Clothes`, `Human Identity`, `Human Interaction`, `Instance Preservation`, `Material`, `Mechanics`, `Motion Order Understanding`, `Motion Rationality`, `Multi-view Consistency`, and `Thermotics`.

We use the augmented text prompts curated for HunyuanVideo and other advanced video diffusion models. Compared to VBench-1.0 used by VideoShield (Hu et al., 2025a), our selected prompt set is more comprehensive, featuring longer and more complex descriptions that better evaluate generative capability. In total, we select 88 text prompts and generate 400 videos for each diffusion model under evaluation. For image-to-video tasks, we employ FLUX-1.0 (Black Forest Labs et al., 2025) to generate image prompts. Examples of text prompts and images used in our experiments are shown in Table 1. The full dataset will be released publicly.

### D.2    Results on Wan-2.2

To further evaluate our approach, we performed additional experiments on the Image-to-Video (I2V) and Text-to-Video (T2V) variants of the Wan-2.2 model. The results demonstrate the significant effectiveness of our method, especially showcasing superior accuracy on the I2V model. We have meticulously recorded the performance across 18 distinct prompt dimensions, with the detailed outcomes presented in Table 2.

## E    Performance bottleneck and future works

Watermarking methods such as VideoShield (Hu et al., 2025a) report nearly $100\%$ bit accuracy, but they are non-blind and incur high extraction costs at scale. In contrast, our method achieves high, though not perfect, extraction accuracy. We attribute this gap to two factors: (1) the imperfect inversion of video diffusion models and (2) the limited error-tolerance of PRC coding.

Figure 1 illustrates the relationship between inversion error and the error-tolerance characteristics of PRC. While PRC introduces diversity by encoding messages into pseudorandom sequences, this

Table 1: Our constructed prompt set – Sample of 5 dimensions with representative prompts

| Dimension | Prompt Text | Image |
|---|---|---|
| **Complex Plot** | The race commenced with explosive energy as the first runner from Team A surged ahead, capturing the attention of all spectators. His swift pace set the tone, but during the baton exchange, a slight fumble nearly cost them. Despite the near mishap, he managed to hand off the baton to the second runner, who, with fierce determination, pursued Team B relentlessly, overtaking them on the curve. However, the earlier error meant their lead was slim. As the third runner took over, nerves were palpable, yet he maintained their position, though Team C's strategic pacing brought them dangerously close. In the final handoff, the pressure was immense. The last runner from Team A accelerated on the penultimate turn, unleashing a powerful sprint that widened the gap. With unwavering focus, he crossed the finish line triumphantly, securing victory for Team A amidst roaring cheers. |  |
| **Composition** | A magnificent lion, its golden mane flowing like a regal crown, possesses the expansive, powerful wings of an eagle, each feather shimmering under the sun's radiant glow. It soars effortlessly through a vast, azure sky, the clouds parting gracefully in its wake. The lion's eyes, sharp and focused, scan the horizon with a kingly gaze, while its muscular body glides with the grace of a seasoned aviator. Below, the earth stretches out in a patchwork of greens and browns, rivers glistening like silver ribbons. The scene captures a breathtaking blend of strength and elegance, as the majestic creature commands the heavens with unparalleled ease. |  |
| **Dynamic Attribute** | In a serene forest, the camera captures a mesmerizing transformation as vibrant red leaves slowly transition to lush green. The scene begins with a close-up of a single leaf, its deep crimson hue glowing under the gentle sunlight. As the camera pans out, the surrounding foliage reveals a tapestry of red, orange, and yellow, creating a warm, autumnal atmosphere. Gradually, the colors shift, with hints of green emerging, symbolizing the renewal of life. The sunlight filters through the canopy, casting dappled shadows on the forest floor, while a gentle breeze rustles the leaves, enhancing the sense of change and rebirth. |  |
| **Dynamic Spatial Relationship** | A playful dog, with its fur slightly ruffled by the breeze, stands on the left side of a wooden dining table, its ears perked up and eyes full of curiosity. The table is surrounded by cozy chairs and a vase with fresh flowers on top. As the dog gets excited, it begins to sprint energetically towards the front of the table, its paws making soft thudding sounds against the wooden floor, eager to reach the other side where a chew toy lies waiting. |  |
| **Instance Preservation** | A vibrant orange dog, with a sleek coat and playful demeanor, sprints joyfully across a sunlit meadow, its ears flapping in the breeze. The dog's eyes sparkle with excitement as it bounds over the lush green grass, leaving a trail of paw prints behind. In the background, wildflowers sway gently, adding bursts of color to the scene. The sun casts a warm glow, highlighting the dog's energetic movements and the natural beauty surrounding it. As the dog runs, its tail wags enthusiastically, embodying pure joy and freedom in the open, expansive landscape. |  |

design compromises its robustness to bit flips. In Figure 1, we plot PRC detection and decoding accuracy (Y-axis) under varying bit-flip rates (X-axis). When the error rate exceeds a certain threshold, PRC decoding accuracy drops sharply toward $0.5$ (random guessing). Moreover, as the encoded message length increases, this threshold becomes lower.

Table 2: Results of different models across different dimensions

| Dimension | Hunyuan-I2V | Hunyuan-T2V | Wan-I2V | Wan-T2V |
|---|---|---|---|---|
| complex plot | 1.0000 | 0.9292 | 1.0000 | 1.0000 |
| composition | 1.0000 | 0.9313 | 1.0000 | 1.0000 |
| dynamic attribute | 1.0000 | 0.8878 | 1.0000 | 0.8974 |
| dynamic spatial relationship | 0.9773 | 0.9991 | 1.0000 | 0.9554 |
| instance preservation | 0.9117 | 0.9584 | 1.0000 | 0.9150 |
| motion rationality | 0.9684 | 1.0000 | 1.0000 | 0.9379 |
| multi-view consistency | 0.9617 | 0.9707 | 0.9500 | 0.8820 |
| camera motion | 0.9560 | 0.9373 | 1.0000 | 1.0000 |
| complex landscape | 0.9913 | 1.0000 | 1.0000 | 1.0000 |
| diversity | 1.0000 | 0.9794 | 1.0000 | 0.9695 |
| human anatomy | 0.9594 | 0.9787 | 1.0000 | 1.0000 |
| human clothes | 0.9879 | 0.9927 | 1.0000 | 0.9744 |
| human identity | 0.9969 | 0.9304 | 1.0000 | 0.9403 |
| human interaction | 1.0000 | 1.0000 | 1.0000 | 0.8864 |
| material | 1.0000 | 0.9738 | 1.0000 | 0.8966 |
| mechanics | 0.9684 | 0.9377 | 1.0000 | 1.0000 |
| motion order understanding | 0.9819 | 0.9617 | 1.0000 | 1.0000 |
| thermotics | 0.9972 | 0.8815 | 0.9500 | 0.9360 |
| Mean acc | 0.9810 | 0.9583 | 0.9944 | 0.9550 |

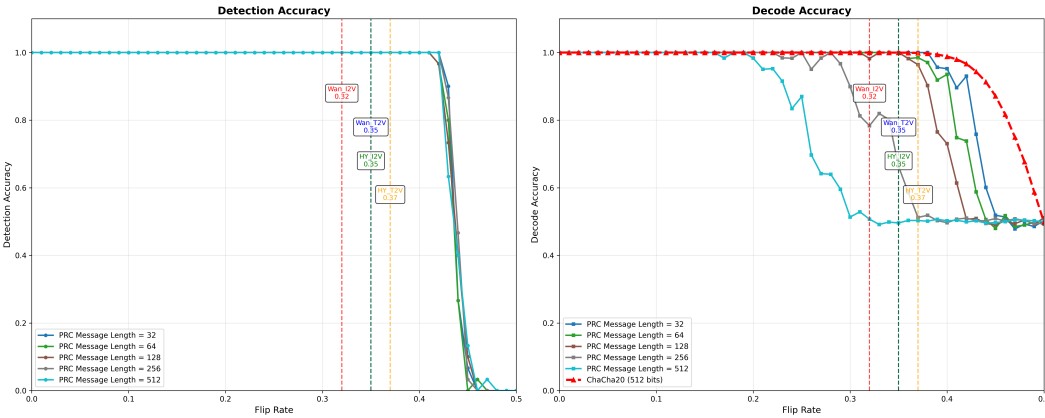

Figure 1: PRC error-tolerance curve and the inversion accuracy of different video diffusion models.

Inversion accuracy for current video diffusion models is far from perfect: HunyuanVideo exhibits an error rate of approximately $0.35$, while Wan-2.2 performs slightly better at $0.32$. When inversion error surpasses PRC's tolerance, decoding becomes effectively random. In contrast, ChaCha20 combined with majority voting (red curve in Figure 1) demonstrates strong error resilience, which explains why non-blind methods like VideoShield achieve near-perfect extraction accuracy.

Fortunately, while PRC.decode() has limited fault tolerance capability, PRC.detect() exhibits strong robustness and has the potential to deliver reliable watermarking under extreme disturbances. We conduct experiments to test the false-positive rate of our proposed SIGMark, feeding the T2V generated videos into I2V model and PRC keys. The results show that 100% of the false-positive samples are distinguished as not generated by I2V model, demonstrating that our method provide reliable watermarking.

Looking forward, we identify two promising directions to improve PRC-based blind watermarking: (1) enhancing inversion accuracy through improved video inversion techniques, and (2) introducing an additional error-correcting layer on top of PRC to mitigate inversion errors.

## F    USAGE OF LARGE LANGUAGE MODELS

This paper utilize Large Language Models (GPT-5) to aid the grammar mistakes and polish writing and formula, and it also helps in dealing with formatting issues during writing in LaTeX. We also use LLM-assisted coding systems in designing and implementing experiments.