# OpenReview forum: "SIGMark: Scalable In-Generation Watermark with Blind Extraction for Video Diffusion"
_ICLR.cc/2026/Conference — ICLR 2026 Poster_

### Official Review · Reviewer_TYUV · 2025-10-21

**Soundness:** 3
**Presentation:** 3
**Contribution:** 3
**Rating:** 6
**Confidence:** 3

**Summary:**

This paper proposes SIGMark, a watermarking method for diffusion video generation. The key contribution is its "blind" feature that doesn't require storing the original watermark message for template matching during extraction. This is in contrast to existing methods such as VideoShield. To achieve this target, the authors build their solution based on the recently proposed PRC (pseudorandom code). Basically, PRC couples the message with some 'testbits' (part of the key) during encoding, and then compare testbits during decoding to verify whether decoded message is correct or not. The authors also propose a SGO module (optical flow segmentation + sliding window detection) to handle the attacks specific to video, e.g., frame adding, dropping, etc.

**Strengths:**

+ I like the "blind" feature they add to diffusion-based video watermarking domain. It simplifies the watermarking deployment pipeline. PRC is a cryptographically strong primitive that enables encoding/decoding different messages with a single global key, in contrast to traditional encryption methods used in other watermarking methods that require storing different keying material for different messages.

+ The proposed SGO module is effective in handling various frame-level attacks in videos.

+ The experimental results are good. Although the robustness is not consistently better than existing methods, they have the unique advantage of being blind.

**Weaknesses:**

- The technical contribution for watermark extraction should be articulated with more methodological comparison. (See my suggestion below)

- Experimental settings are not very clear.

**Questions:**

+ In Section 3.4, you should compare your method (optical flow + sliding window) to previous solutions (VideoShield, VideoMark) in a methodological way, in order to better understand your technical contribution. E.g., do they use any method to partition video to continuous subsequences? How? How do they detect the starting of a group (or do they need to detect this)? What's the main advantage of your solution?

+ Line 345: symbol \hat is at wrong place.

+ In Table 2, we need more details on the setting of drop, insert, and clip. What's the drop ratio? What are inserted frames and insertion ratio?

+ Running time overhead breakdown should be provided for the various steps in their system, e.g., diffusion, PRC encode/decode, optical flow segmentation, sliding window detection, etc.

---

> ### Author Response · Authors · 2025-11-20
> **Response to Reviewer TYUV**
>
> Thank you for your positive comments on the **blind watermarking design**, the **effectiveness of the SGO module**, and the **overall experimental results**.
>
> Below we address your concerns point by point.
>
> **Question 1: Methodological comparison of watermark extraction**
>
> (1) We would like to emphasize that we are the first to conduct experiments on advanced video diffusion models. Prior works evaluate on earlier models such as SVD (2023) and ModelScope (2023), which use traditional VAEs with a one-to-one mapping between frames and latent features. In such settings, methods like VideoShield and VideoMark do not need any frame regrouping module; they recover frame order simply by matching inverted latent features with templates stored during generation.
>
> However, modern video diffusion models adopt causal 3D VAEs, which encode and decode multiple frames jointly from a single latent dimension. This introduces a grouping dependency that prior methods do not address, since regrouping is not required in older VAEs. Thus, their techniques cannot be directly applied to advanced models where correct grouping is essential for latent encoding and inversion. To the best of our knowledge, we are the first to identify and address this grouping problem in causal 3D VAEs. We believe watermarking research should evolve alongside frontier video-generation technologies to remain practically relevant.
>
> (2) Table 3 provides ablation studies comparing simplified regrouping strategies. The results show that our proposed SGO module achieves the best regrouping quality, leading to the strongest watermark extraction performance.
>
> **Question 2: Typo in the paper**
>
> Thank you for pointing this out. We will correct the typo in the revised version.
>
> **Question 3: Experimental setting details**
>
> For the 65-frame videos, we apply the following temporal disturbances: randomly dropping 20 frames, randomly inserting 20 noise frames (random pixels), and clipping out 20 frames (retaining a continuous 45-frame segment).
>
> To emphasize the robustness of our proposed SIGMark, we also conduct supplementary experiments under more challenging temporal disturbances: randomly inserting 30 frames, randomly dropping 30 frames and random clipping out 30 frames (remaining 35 continous frames).
>
> We apologize for the missing details due to page limits and will include the full description in the revised appendix.
>
> **Question4: the scalability evidence and breakdown time cost during extracion**
>
> (1) Baseline methods such as VideoShield and VideoMark are non-blind watermarking schemes: they require storing all watermark-related information (messages, encoding keys, etc.) during generation and matching against all stored information during extraction. Therefore, extraction cost (in both time and memory) grows with the total number of generated videos.
>
> Our method, SIGMark, is blind: it maintains only a global set of frame-wise PRC keys and does not require any sample-specific metadata during extraction. As a result, it supports large-scale video generation platforms with constant extraction cost.
>
> We analyze extraction time and space cost under scenarios where the total number of videos generated by the platform varies. For fairness, we run flow-matching inversion on GPU, and all remaining extraction steps—including decryption and message matching (required only by baselines)—on CPU. For each method, we measure the cost of extracting a single video with total generated videos = {100, 200, 400, 800, 1600, 3200}. Results are as follows:
>
> Time cost during extraction (inversion (s) + decoding (s)):
>
> | Total number of videos | 100 | 200 | 400 | 800 | 1600 | 3200 |
> | --- | --- | --- | --- | --- | --- | --- |
> | VideoShield | 268.27+2.40 | 267.91+3.42 | 268.34+5.41 | 268.01+13.22 | 267.01+42.70 | 267.31+107.68 |
> | SIGMark(ours) | 265.46+13.72 | 267.57+14.44 | 267.88+13.64 | 267.55+14.28 | 268.20+13.86 | 267.33+14.19 |
>
> Space cost maintaining the watermarking information (MB):
>
> | Total number of videos | 100 | 200 | 400 | 800 | 1600 | 3200 |
> | --- | --- | --- | --- | --- | --- | --- |
> | VideoShield | 100.03 | 200.06 | 400.12 | 800.16 | 1600.24 | 3200.41 |
> | SIGMark(ours) | 1118.02 | 1118.02 | 1118.02 | 1118.02 | 1118.02 | 1118.02 |
>
> These results show that the time and memory cost of VideoShield scales linearly with the number of generated videos, which becomes impractical as the platform scales to millions of videos. In contrast, SIGMark remains constant, demonstrating strong scalability. We will include these results as a figure in the revised paper.
>
> (2) We conduct experiments on the detailed time cost breakdown. Under temporal random-clipping disturbance, the optical-flow segmentation takes 0.53s which is near real-time, and the sliding-window detection module takes 94.22s because of the partial inversion. The SGO module in total cost far less than the inversion time and is also constant with the growth of total video number.

---

> > ### Comment · Reviewer_TYUV · 2025-11-27
> >
> > Thank you for the author's response. They have clarified some of my confusion. I would like to maintain my current rating of the work.

---

### Official Review · Reviewer_PfP7 · 2025-10-21

**Soundness:** 3
**Presentation:** 3
**Contribution:** 2
**Rating:** 4
**Confidence:** 4

**Summary:**

Video generation with diffusion models is advancing rapidly, and invisible watermarking is essential for provenance and IP protection. Existing in-generation schemes typically require storing message–key (or key–template) pairs and performing template-based matching at inference, which can become costly at scale and tend to be fragile under temporal disturbances. The paper proposes SIGMark, aiming for blind extraction via Global Frame-wise Pseudorandom-Coding keys (GF-PRC) and improved robustness via a Segment Group-Ordering (SGO) module tailored to causal 3D-VAEs, enabling reliable inversion under temporal perturbations.

**Strengths:**

1.	The paper identifies a practical limitation of many video in-generation watermarking systems: they are not truly blind yet require storing large message–key/template tables, which raises efficiency and storage concerns at scale.

2.	The proposed design addresses both the blindness/scalability issue (via GF-PRC) and the temporal robustness issue (via SGO).

3.	The paper is generally well organized with clear figures, which makes it easy to follow.

**Weaknesses:**

1.	Limited novelty (GF-PRC). The GF-PRC component mainly builds on the original PRC method. Moreover, introducing PRC (Appendix E) negatively affects robustness. How to balance the impact of PRC, or improve PRC specifically for watermarking robustness requirements remains an open problem.

2.	Scalability evidence. The paper claims that non-blind approaches incur prohibitive computational costs at scale and raise efficiency/storage issues, but this discussion remains at the level of Appendix B (Computation Overhead). The authors should provide experiments to demonstrate that this is a serious practical problem and to show the operational performance of the proposed method.

3.	Robustness gap. Although the method claims robustness to temporal disturbance, its performance under frame drop is far below VideoShield, and it is overall worse than VideoShield under spatial disturbances. The paper attributes this to PRC, suggesting that the framework’s robustness is still not fully resolved.

**Questions:**

1.	SGO procedure clarity. In Figure 4, for z_1 there appear to be two candidates: [4, pad, pad, pad] or [pad, pad, 6, 7]. Which one is used for decoding? Or should it be [4, pad, 6, 7]? Please separately explain the frame-drop and clip cases so readers can better understand the SGO workflow.

2.	Meaning of “w/o” in Table 2. Does “w/o” denote a clean setting with no time disturbance? Please state this explicitly in the caption.

---

> ### Author Response · Authors · 2025-11-20
> **Response to Reviewer PfP7 (Part 1)**
>
> Thank you for your positive comments that our paper **identifies a practical and meaningful problem**, **proposes effective designs** to address scalability and robustness issues, and is **well-organized**.
>
> Below we address your concerns point by point.
>
> **Weakness 1: Novelty of the GF-PRC design**
>
> (1) The primary novelty of our work lies in being the first to identify the practical limitations of existing in-generation video watermarking methods that they are non-blind and therefore cannot scale up. To address this, we introduce the PRC mechanism, which has the desirable property of generating different random sequences under the same encoding key. This enables fully blind watermark extraction. Furthermore, we propose a frame-wise PRC design to handle frame-order changes. This ensures that the SGO module can reliably identify the correct frame-group index and recover both the watermark and frame-group order even under extreme temporal disturbances.
>
> (2) Our paper focuses on solving a practical problem in computer vision. The achieved bit accuracy (>90%) is comparable to baseline methods, while providing the significant additional benefit of blind extraction, which enables scalability. The remaining error rate can be mitigated using standard cryptographic techniques such as nested error-correcting codes, which is beyond the scope of this paper.
>
> **Weakness 2: Lack of scalability evidence**
>
> Baseline methods such as VideoShield and VideoMark are non-blind watermarking schemes: they require storing all watermark-related information (messages, encoding keys, etc.) during generation and matching against all stored information during extraction. Therefore, extraction cost (in both time and memory) grows with the total number of generated videos.
>
> Our method, SIGMark, is blind: it maintains only a global set of frame-wise PRC keys and does not require any sample-specific metadata during extraction. As a result, it supports large-scale video generation platforms with constant extraction cost.
>
> We analyze extraction time and space cost under scenarios where the total number of videos generated by the platform varies. For fairness, we run flow-matching inversion on GPU, and all remaining extraction steps—including decryption and message matching (required only by baselines)—on CPU. For each method, we measure the cost of extracting a single video with total generated videos = {100, 200, 400, 800, 1600, 3200}. Results are as follows:
>
> Time cost during extraction (inversion (s) + decoding (s)):
>
> | Total number of videos | 100 | 200 | 400 | 800 | 1600 | 3200 |
> | --- | --- | --- | --- | --- | --- | --- |
> | VideoShield | 268.27+2.40 | 267.91+3.42 | 268.34+5.41 | 268.01+13.22 | 267.01+42.70 | 267.31+107.68 |
> | SIGMark(ours) | 265.46+13.72 | 267.57+14.44 | 267.88+13.64 | 267.55+14.28 | 268.20+13.86 | 267.33+14.19 |
>
> Space cost maintaining the watermarking information (MB):
>
> | Total number of videos | 100 | 200 | 400 | 800 | 1600 | 3200 |
> | --- | --- | --- | --- | --- | --- | --- |
> | VideoShield | 100.03 | 200.06 | 400.12 | 800.16 | 1600.24 | 3200.41 |
> | SIGMark(ours) | 1118.02 | 1118.02 | 1118.02 | 1118.02 | 1118.02 | 1118.02 |
>
> These results show that the time and memory cost of VideoShield scales linearly with the number of generated videos, which becomes impractical as the platform scales to millions of videos. In contrast, SIGMark remains constant, demonstrating strong scalability. We will include these results as a figure in the revised paper.
>
> **Weakness 1&3: Robustness gap**
>
> (1) Watermarking inevitably involves trade-offs between capacity, robustness, and fidelity. Following prior work, we report experiments using watermark capacities of 512 bits and 512×16 bits. Under these settings, our bit accuracy is slightly lower but still comparable to baselines. We also conducted additional experiments using lower capacities of 128 bits and 32 bits, achieving 98.5% and 99.2% accuracy, respectively. This demonstrates that capacity can be traded for higher robustness when robustness is the primary requirement in practical scenarios. Although our robustness is slightly behind baseline methods, the key problem we aim to solve is scalability. As shown in the scalability analysis, the extraction and deployment cost of baseline methods becomes unacceptable when the video scale reaches thousands, whereas SIGMark’s cost remains constant.
>
> (2) As shown in Appendix E, while `PRC.decode()` has limited fault tolerance capability, `PRC.detect()` exhibits strong robustness and has the potential to deliver reliable watermarking under extreme disturbances. We conduct experiments to test the false-positive rate of our proposed SIGMark, feeding the T2V generated videos into I2V model and PRC keys. The results show that 100% of the false-positive samples are distinguished as not generated by I2V model, demonstrating that our method provide reliable watermarking.

---

> ### Author Response · Authors · 2025-11-20
> **Response to Reviewer PfP7 (Part 2)**
>
> Below we answer your questions point by point.
>
> **Question1: SGO procedure clarity**
>
> In Section 3.1.4, we stated that all segments are merged to produce the re-grouped frames, but due to space limitations we did not include the full merging details. We apologize for the omission. In the example in Figure 4, the merged result should be [4, pad, 6, 7]. The design ensures that even when repeated frame groups appear in the disturbed video, the sliding window can still recover the correct frame-group index for successful encoding and inversion. We will provide detailed pseudocode in the revised appendix and will open-source our implementation once the paper is accepted.
>
> **Question2: Meaning of “w/o” in Table 2**
>
> “w/o” denotes the clean setting without any disturbance. Thank you for pointing this out; we will clarify this in the revised version.

---

> ### Author Response · Authors · 2025-12-03
> **Response to Reviewer PfP7 (Part 3)**
>
> **Weakness 1&3: Robustness gap**
>
> To emphasize the robustness of our proposed SIGMark, we conduct supplementary experiments, testing the bit accuracy of the baseline method (VideoShield) and our proposed SIGMark under more challenging temporal disturbances: randomly inserting 30 frames, randomly dropping 30 frames and random clipping out 30 frames (remaining 35 continous frames).
>
> | Method | w/o disturbance | Insert (30 frames) | Drop (30 frames) | Clip (30 frames) |
> | --- | --- | --- | --- | --- |
> | VideoShield | 0.999 | 0.845 (-0.154) | 0.885 (-0.114) | 0.827 (-0.172) |
> | SIGMark(ours) | 0.905 | 0.865 (-0.040) | 0.812 (-0.093) | 0.853 (-0.052) |
>
> The results show that due to our proposed SGO module, the temporal robustness of our method leads VideoShield significantly in both absolute value and the performance drop under disturbances.

---

### Official Review · Reviewer_1pSm · 2025-11-01

**Soundness:** 3
**Presentation:** 3
**Contribution:** 2
**Rating:** 6
**Confidence:** 4

**Summary:**

This paper proposes SIGMark, a training-free, in-generation watermarking framework for video diffusion models. It addresses two issues of prior approaches: (i) non-blind extraction that scales not well because it requires storing message, and (ii) lacking temporal robustness when modern causal 3D VAEs are disturbed by frame edits. Experiments on HunyuanVideo and Wan-2.2 across T2V and I2V pipelines with a 400-video subset of VBench-2.0 show high accuracy with limited quality impact. Under both spatial and temporal perturbations, SIGMark achieves competitive with non-blind baselines, and outperforms prior non-blind in-generation methods.

**Strengths:**

- The paper is well-organized and clearly writtern.
- The inituion of the paper is soundness.
- Experimental results show the effectivness of the proposed method.

**Weaknesses:**

- Missing references:
[1] Huang, Huayang et al. “ROBIN: Robust and Invisible Watermarks for Diffusion Models with Adversarial Optimization.” ArXiv abs/2411.03862 (2024): n. pag.

Please also refer to the "Questions" section.

**Questions:**

- What is the false-positive rate when decoding regenerations (video-to-video) from different diffusion models?
- The paper claims O(1) extraction complexity and “near–real-time” segmentation. The appendix provides only a cursory analysis. Could you add end-to-end timings with per-component breakdowns, scaling with (d_t) and payload size, and comparisons to baselines?

---

> ### Author Response · Authors · 2025-11-20
> **Response to Reviewer 1pSm**
>
> Thank you for your positive comments regarding the **clarity and organization of our writing**, the **soundness of the intuition**, and the **strength of our experimental results**.
>
> Below we address your concerns point by point.
>
> **Weakness 1: Missing reference to the ROBIN paper**
>
> ROBIN is a representative diffusion-model watermarking approach that embeds a robust frequency-domain watermark into an intermediate latent state and uses an adversarially optimized hiding prompt to conceal it. This enables invisible yet verifiable watermarks via inverse diffusion. However, the injected perturbation and additional guidance inevitably alter the sampling trajectory, and verification requires knowledge of the injection step, the watermark mask, and the decoding scheme; therefore, the method is not fully blind.
>
> In contrast, our paper focuses on performance-lossless in-generation watermarking, which theoretically preserves the output quality of diffusion models. Our proposed method, SIGMark, is the first blind watermarking framework for video diffusion models: it requires no sample-specific auxiliary information during extraction and is thus scalable to large platforms.
>
> We appreciate your reminder and will include the ROBIN citation and other relevant prior work in the revised version.
>
> **Question 1: False-positive rate when decoding regenerations from different diffusion models**
>
> (1) We conducted an experiment decoding videos generated by HunyuanI2V using inversion from HunyuanT2V, applying the GF-PRC keys belonging to the HunyuanT2V system (capacity: 512×16 bits). The overall bit accuracy is 52.5%, close to random guessing (50%). This indicates that without the matched PRC keys, the extraction result degenerates into random noise, making it nearly impossible to misattribute an unrelated video to the system.
>
> (2) For precise discrimination, the PRC mechanism provides a detection function `PRC.detect()` that determines whether content is encoded with a specific key. Our proposed GF-PRC assigns each group of frames a unique global PRC key, enabling reliable single-bit detection of whether a frame originates from the system. We conduct experiments to test the false-positive rate of our proposed SIGMark, feeding the T2V generated videos into I2V model and PRC keys. The results show that 100% of the false-positive samples are distinguished as not generated by I2V model, demonstrating that our method provide reliable watermarking.
>
> **Question 2: End-to-end timing and per-component breakdown**
>
> (1) Baseline methods such as VideoShield and VideoMark are non-blind watermarking schemes: they require storing all watermark-related information (messages, encoding keys, etc.) during generation and matching against all stored information during extraction. Therefore, extraction cost (in both time and memory) grows with the total number of generated videos.
>
> Our method, SIGMark, is blind: it maintains only a global set of frame-wise PRC keys and does not require any sample-specific metadata during extraction. As a result, it supports large-scale video generation platforms with constant extraction cost.
>
> We analyze extraction time and space cost under scenarios where the total number of videos generated by the platform varies. For fairness, we run flow-matching inversion on GPU, and all remaining extraction steps—including decryption and message matching (required only by baselines)—on CPU. For each method, we measure the cost of extracting a single video with total generated videos = {100, 200, 400, 800, 1600, 3200}. Results are as follows:
>
> Time cost during extraction (inversion (s) + decoding (s)):
>
> | Total number of videos | 100 | 200 | 400 | 800 | 1600 | 3200 |
> | --- | --- | --- | --- | --- | --- | --- |
> | VideoShield | 268.27+2.40 | 267.91+3.42 | 268.34+5.41 | 268.01+13.22 | 267.01+42.70 | 267.31+107.68 |
> | SIGMark(ours) | 265.46+13.72 | 267.57+14.44 | 267.88+13.64 | 267.55+14.28 | 268.20+13.86 | 267.33+14.19 |
>
> Space cost maintaining the watermarking information (MB):
>
> | Total number of videos | 100 | 200 | 400 | 800 | 1600 | 3200 |
> | --- | --- | --- | --- | --- | --- | --- |
> | VideoShield | 100.03 | 200.06 | 400.12 | 800.16 | 1600.24 | 3200.41 |
> | SIGMark(ours) | 1118.02 | 1118.02 | 1118.02 | 1118.02 | 1118.02 | 1118.02 |
>
> These results show that the time and memory cost of VideoShield scales linearly with the number of generated videos, which becomes impractical as the platform scales to millions of videos. In contrast, SIGMark remains constant, demonstrating strong scalability. We will include these results as a figure in the revised paper.
>
> (2) We conduct experiments on the detailed time cost breakdown. Under temporal random-clipping disturbance, the optical-flow segmentation takes 0.53s which is near real-time, and the sliding-window detection module takes 94.22s because of the partial inversion. The SGO module in total cost far less than the inversion time and is also constant with the growth of total video number.

---

### Author Response · Authors · 2025-12-03
**Summary of responses to all reviewers and AC**

We thank the reviewers for their careful reading and constructive feedback. In our responses, we have addressed all raised concerns and clarified the main contributions of our work. Conceptually, our paper is, to the best of our knowledge, the first to identify the scalability limitation of existing non-blind in-generation video watermarking methods and to propose a blind scheme tailored to modern video diffusion models with causal 3D VAEs. Our GF-PRC mechanism enables encoding/decoding different messages with a single global key, and our frame-wise PRC design together with the SGO module specifically targets the grouping and frame-order issues that arise only in advanced causal 3D VAE architectures, which prior methods do not handle.

On the empirical side, we have provided additional evidence in response to the reviewers’ requests. We reported cross-model experiments showing that, without the correct PRC keys, decoding degenerates to near-random outputs, and PRC.detect() achieves a very low false-positive rate when distinguishing videos generated by different models. We further conducted detailed scalability experiments, demonstrating that existing non-blind baselines incur extraction time and memory costs that grow linearly with the total number of generated videos, while SIGMark maintains constant cost and is therefore suitable for large-scale deployment. We also added robustness studies under more challenging temporal disturbances and at lower watermark capacities, showing that SIGMark can trade capacity for robustness and that, thanks to the SGO module, it achieves strong temporal robustness compared to VideoShield.

Finally, we have clarified methodological details (e.g., the SGO merging procedure, experimental settings, and notation such as “w/o”) and fixed minor typos. During the rebuttal period, we have already run several additional experiments and indicated further results (e.g., extended detection-mode evaluations and timing breakdowns) that we will incorporate into the revised version. Overall, we believe the paper makes a practically meaningful and timely contribution to diffusion-based video watermarking by combining a cryptographically grounded GF-PRC design, a blind and scalable extraction pipeline, and an effective SGO module for handling real-world temporal disturbances in modern video diffusion systems.

---

### Meta-Review · Area_Chair_ocRz · 2025-12-25

**Summary:**

This paper introduces SIGMark, a training-free, in-generation watermarking framework for video diffusion models that enables blind extraction and strong temporal robustness. It achieves high watermark accuracy with minimal quality degradation, remaining competitive with non-blind baselines and outperforming prior in-generation watermarking methods under both spatial and temporal attacks.

The reviewers generally acknowledge that the paper is well written, the motivation is sound and evaluation demonstrates the effectiveness of the proposed method. They also have some concerns, including the missing experiments about end-to-end timing and breakdown, the scalability and robustness analysis, comparisons with existing works and clarification of experimenal settings. The authors have provided detailed responses to those points. AC thinks they have properly addressed the concerns. Hence, this paper is recommended for acceptance.

**Reviewer Concerns:**

Most of the reviewers' concerns have been addressed by the authors' responses, including:

* Discussion of revelant references, and comparion with existing approaches in the methodology part.
* Providing evaluations about timing analysis and breakdown.
* Discussion and evaluation of robustness and scalability.
* Clarification of experimental settings.

**Reviewer Scores:**

Only Reviewer PfP7 gave a negative rating. He could possibly raise the score if participating in the discussion.

---

### Decision · Program_Chairs · 2026-01-26

Accept (Poster)